# Preoperative Molecular Subtype Classification Prediction of Ovarian Cancer Based on Multi-Parametric Magnetic Resonance Imaging Multi-Sequence Feature Fusion Network

**DOI:** 10.3390/bioengineering11050472

**Published:** 2024-05-09

**Authors:** Yijiang Du, Tingting Wang, Linhao Qu, Haiming Li, Qinhao Guo, Haoran Wang, Xinyuan Liu, Xiaohua Wu, Zhijian Song

**Affiliations:** 1Digital Medical Research Center, School of Basic Medical Sciences, Fudan University, Shanghai 200032, China; 22111010028@m.fudan.edu.cn (Y.D.);; 2Shanghai Key Laboratory of Medical Imaging Computing and Computer Assisted Intervention, Shanghai 200032, China; 3Department of Nuclear Medicine, Ren Ji Hospital, Shanghai Jiao Tong University School of Medicine, Shanghai 200127, China; 4Department of Gynecologic Oncology, Fudan University Shanghai Cancer Center, Fudan University, Shanghai 200032, China; 5Department of Oncology, Shanghai Medical College, Fudan University, Shanghai 200032, China

**Keywords:** ovarian cancer, multi-parametric magnetic resonance imaging, deep learning

## Abstract

In the study of the deep learning classification of medical images, deep learning models are applied to analyze images, aiming to achieve the goals of assisting diagnosis and preoperative assessment. Currently, most research classifies and predicts normal and cancer cells by inputting single-parameter images into trained models. However, for ovarian cancer (OC), identifying its different subtypes is crucial for predicting disease prognosis. In particular, the need to distinguish high-grade serous carcinoma from clear cell carcinoma preoperatively through non-invasive means has not been fully addressed. This study proposes a deep learning (DL) method based on the fusion of multi-parametric magnetic resonance imaging (mpMRI) data, aimed at improving the accuracy of preoperative ovarian cancer subtype classification. By constructing a new deep learning network architecture that integrates various sequence features, this architecture achieves the high-precision prediction of the typing of high-grade serous carcinoma and clear cell carcinoma, achieving an AUC of 91.62% and an AP of 95.13% in the classification of ovarian cancer subtypes.

## 1. Introduction

Ovarian cancer (OC) occupies a significant position among female malignancies, representing an important challenge in the public health field due to its high mortality rate and the complexity of diagnosis and treatment. the overall five-year survival rate for ovarian cancer is between 30% to 40% [1]. However, since ovarian cancer usually shows no symptoms in its early stages, the survival rate for stage 4 ovarian cancer drops to as low as 3% [2], reflecting the high lethality of ovarian cancer. Due to the high heterogeneity of ovarian cancer, encompassing various histological subtypes such as high-grade serous carcinoma and clear cell carcinoma, accurate preoperative subtype determination is crucial for guiding treatment choices and predicting treatment efficacy [3]. For example, clear cell carcinoma in ovarian cancer, which has an indolent course and is predominantly platinum-resistant, leads clinicians to prefer a “surgery-first, chemotherapy-second” approach; whereas high-grade serous carcinoma, known for its aggressiveness and sensitivity to chemotherapy, leads to a “chemotherapy-first, surgery-second” preference. However, traditional subtype identification relies mainly on invasive intraoperative pathological biopsy procedures [4], which not only carry a certain risk of iatrogenic tumor dissemination but also face limitations due to tumor location or patient conditions making biopsy unfeasible.The development of imaging technology, especially multi-parametric magnetic resonance imaging (mpMRI), provides a new means for the non-invasive assessment of ovarian cancer typing. mpMRI can detail the morphological features and tissue structure of tumors [5,6,7,8,9,10,11]. However, the accurate diagnosis of ovarian cancer typing by MRI requires extensive expertise and is subject to the subjective influence of the doctor’s experience [12,13,14,15,16].

The development of deep learning (DL) technology has had a profound impact in the field of medical image analysis [17,18,19,20,21,22,23,24,25,26,27,28,29,30]. Through its powerful data processing and feature recognition capabilities, deep learning can automatically extract and learn valuable information from complex MRI data, recognizing subtle patterns and variations that might be overlooked by human experts, providing new opportunities for the automated analysis of complex MRI data [31,32]. Recent advancements in convolutional neural networks (CNN) and similar deep learning models have made a significant impact in the field of medical diagnostics. Kott et al. [33] used deep residual CNNs for the histopathological diagnosis of prostate cancer, demonstrating the model’s capability to roughly classify image blocks as benign or malignant. Ismael et al. [34] proposed a method using residual networks (ResNet50 architecture) for automatic brain tumor classification, proving its effectiveness at the patient level. Booma et al. [35] introduced a method enhanced with ML algorithms and max pooling, achieving an accuracy rate of 89%. Wen et al. [36] utilized a custom set of 3D filters, with accuracies ranging from 83% to 90%. Wang et al. [37] proposed a two-stage deep transfer learning method, reaching an accuracy of 87.54%. Despite the widespread application of deep learning in medical diagnostic tasks, research on classifying subtypes of ovarian cancer is rare. Most studies focus on classifying a single subtype of ovarian cancer as negative or positive [38,39,40,41,42]. However, ovarian cancer is not a single disease but a group of diseases with different biological characteristics, treatment responses, and prognoses. Subtyping, as opposed to classifying a single subtype, provides a greater volume of information and presents a higher difficulty level. In recent years, EfficientNet [43], supported by Neural Architecture Search (NAS) [44], maintains a balance between classification performance and model size, and extensive testing has shown the good generalizability of this network structure. This progress not only highlights the application prospects of deep learning technology in the medical field but also shows the possibility of achieving a balance between model efficiency and performance through advanced algorithm optimization. To explore the feasibility of this approach with real clinical data, this paper proposes an end-to-end preoperative diagnostic model for high-grade serous carcinoma and clear cell carcinoma of ovarian cancer, based on EfficientNet and multi-parametric MRI sequence feature fusion. Initially, we use the EfficientNet feature extractor to independently extract features from each parametric sequence, ensuring the globality of the extracted features. Secondly, we propose a feature fusion strategy to fully utilize and integrate the comprehensive information of T1- and T2-weighted MRI images, combining the complementary information from different sequences. Lastly, the model we developed can automatically complete high-precision subtype prediction end-to-end, making it more conducive to clinical application.

In our study, we constructed a comprehensive clinical dataset, collecting data from 311 patients, of which 250 were used for training and validation, and 61 for independent testing. All patients included enhanced T1 and T2 sequences, and the subtypes were diagnosed by intraoperative pathological tissue biopsy. To evaluate the performance of the built model, we used metrics such as the area under the receiver operating characteristic (ROC) curve (AUC), accuracy, sensitivity, and specificity to assess the model’s efficacy. The experimental results demonstrate the potential application of our deep learning model in the preoperative non-invasive subtype assessment of ovarian cancer. Through the fusion of multi-sequence MRI features, our model not only achieved high-accuracy prediction but also provided a safer, more accurate, and less subjective new mode of automatic diagnosis for ovarian cancer. The structure of this paper is as follows: The Section 2 will present the materials and methods, including data sources, data preprocessing, model architecture, and parameter details. The Section 3 will analyze experimental results and discuss model parameter selection. The Section 4 will discuss the findings and limitations of the study. Finally, the Section 5 will summarize the research and explore its significance and potential application prospects in related fields.

## 2. Materials and Methods

This study included 311 patient samples, of which 248 were used for model training and validation, and the remaining 63 were used for independent testing. All patient samples contained enhanced T1 sequence and T2 sequence images, and all typing results were confirmed by intraoperative pathological biopsy. A multi-parametric feature fusion model was constructed for this experiment. First, we used the EfficientNet feature extractor to independently extract features from each sequence. Secondly, we fused the features of MRI’s T1 and T2, combining information from different sequences. Finally, our system was capable of end-to-end prediction of ovarian cancer subtype classification, with the input being the MRI image sequences of T1 and T2, and the output being the predicted classification of ovarian cancer for the patient.

### 2.1. Data Sources

In this study, we included patients from Fudan University Cancer Hospital preliminarily suspected of having epithelial ovarian cancer, who underwent MRI examinations before surgery. This study was approved by the relevant ethical review board. Figure 1 provides a detailed demonstration of the sources, screening, and division of data in this study. The inclusion criteria for patients were as follows: no contraindications for MRI examination, surgery, and histopathology confirmed high-grade serous ovarian cancer/ovarian clear cell carcinoma (HGSOC/OCCC), underwent a pre-treatment pelvic MRI examination, and no history of chemotherapy. Based on these criteria, we excluded some patients: those who received neoadjuvant chemotherapy before initial debulking surgery, those confirmed as early-stage HGSOC/OCCC, lost to follow-up patients, those who did not receive standardized chemotherapy, those confirmed as peritoneal HGSOC/OCCC, and those with poor image quality or a lack of solid components. Ultimately, a total of 311 patients were included in the study. These patients were then allocated to different datasets: 198 patients were assigned to the training dataset, 50 to the validation dataset, and 63 to the test dataset. This division was made to evaluate the performance of the developed models and ensure the models have a good generalizability across various datasets.

### 2.2. MRI Image Preprocessing

Preprocessing in deep learning is the step that involves preprocessing the input data in order to make the data more suitable for model training, thereby improving the efficiency and performance of the model. The preprocessing of the original ovarian MRI images of patients includes images of high-grade serous carcinoma (212 images) and clear cell carcinoma (99 images), all containing two modalities: T1 with contrast enhancement (T1+C, hereinafter referred to as T1) and T2-weighted imaging (T2WI, hereinafter referred to as T2). T1-weighted contrast-enhanced imaging (T1+C) involves imaging after intravenous injection of a contrast agent on the basis of T1-weighted imaging. This imaging technique is suitable for detecting vascular-rich tissues and tumors because they become more prominent in the images due to the absorption of the contrast agent. T2-weighted imaging (T2WI) provides information different from T1WI, mainly highlighting tissues with a high water content. T2 is particularly useful for observing fluids and distinguishing between cystic and solid lesions. In the diagnosis of ovarian cancer, T1 and T2 provide different information: T1 helps assess the tumor’s angiogenesis and boundaries; T2 helps identify the nature of the tumor (such as whether it contains cystic components) and differentiate the tumor from surrounding tissues. The specific steps are as follows: (1) Exclude other modalities, leaving only T1+C (hereinafter referred to as T1) and T2WI (hereinafter referred to as T2); (2) convert all files to NIfTI format; (3) randomly select one patient’s T1 image as a template to register all other T1 modality images, aligning all images in size and space, using correlation as the registration objective function; (4) align all patients’ T2 images in size and space with the template T1 image through cross-modal registration, with the objective function for multimodal registration being Mutual Information. The images before and after preprocessing are shown in Figure 2, with T1 and the registered T2 serving as model inputs after aligning the T2 image to the same size as T1.

### 2.3. Ovarian Cancer Classification Prediction Model Based on mpMRI and EfficientNet Multi-Sequence Feature Fusion

Figure 3 provides an overview of the methodology used in this paper, utilizing two MRI sequences (T1 and T2) from the initial examination of patients. This method includes three stages: (1) Deep feature extraction: at this stage, MRI images of ovarian cancer T1 and T2 are input into an EfficientNet-based feature extractor to obtain features corresponding to each patient’s MRI sequence; (2) feature fusion: this stage first aggregates the deep features generated from all sequences of the same patient to obtain fused features; (3) ovarian cancer subtype prediction: at this stage, deep features are passed through a fully connected layer to obtain predictions for ovarian cancer subtypes. It is important to note whether it is a single sequence or a fused sequence, as deep features have corresponding prediction branches, allowing for both single sequence and fused sequence ovarian cancer subtype predictions. As shown in Figure 3, if a patient’s initial examination is missing a sequence, this method can still provide prediction results for other sequences; hence, the method is robust.

### 2.4. Deep Feature Extraction Based on EfficientNet

EfficientNet [43] utilizes Neural Architecture Search (NAS) technology to find an optimal configuration of three parameters: the network’s image input resolution, depth, and channel width, to achieve higher predictive accuracy. This led to the development of EfficientNet-B0 as the base model. Subsequently, through Compound Scaling, it systematically scales the input image resolution, depth, and width of the base model, generating a series of larger models: EfficientNetB1-B7. Through ablation studies, EfficientNet-B2 was ultimately chosen as the deep feature extractor. The ablation studies will be discussed in detail in Section 4 of the article. Its network structure is shown in Table 1, where Conv3 × 3 denotes a conventional convolutional layer with a kernel size of 3 × 3, MBConv1 indicates an expansion factor of 1, MBConv6 indicates an expansion factor of 6, and k3 × 3 denotes the kernel size of 3 × 3 for layer-by-layer convolution in the corresponding module. It is important to note that the weights of the EfficientNet feature extractor for each sequence are different, with each EfficientNet feature extractor only extracting features from its corresponding sequence. Using the EfficientNet feature extractor enhances network performance through increased width, depth, and input resolution. Increasing the number of convolutional kernels (width) improves feature granularity and eases training. Adding more layers (depth) facilitates the capture of richer and more complex features. Enhancing the input resolution can yield higher granularity in feature templates.

### 2.5. Multi-Sequence Feature Fusion

This study employs two MRI sequences, and the approach to multi-sequence feature fusion significantly influences the prediction outcomes. We explore two primary fusion strategies: Concatenate and Add. The Concatenate method joins feature vectors from various sources end-to-end, forming an extended vector with no information loss and straightforward implementation. Conversely, the Add method executes element-wise addition on feature vectors, necessitating uniform dimensions across these vectors. Through ablation studies, as detailed in Section 4, we selected the Concatenate method for our multi-sequence feature fusion.

Given a data-augmented sequence *j* for patient *i* denoted as $x_{i}^{j}$, and utilizing the EfficientNet feature extractor for each sequence represented by Feat(), the deep features for $x_{i}^{j}$ are expressed as $f_{i}^{j}=Feat\left(x_{i}^{j}\right)$. The fusion via concatenation is formulated as: (1)$$f_{i}=\mathrm{Concat}\left(\left[f_{i}^{1},f_{i}^{2}\right]\right),$$
where $f_{i}$ represents the fused feature vector of sample *i*. This fusion approach integrates features from both the T1 and T2 MRI sequences, denoted as $f_{i}^{1}$ (T1 features) and $f_{i}^{2}$ (T2 features), respectively. Following this fusion, three distinct feature sets are obtained: features from the T1 sequence, the T2 sequence, and the combined features $f_{i}$.

### 2.6. Ovarian Cancer Subtype Prediction

After obtaining the fused feature $f_{i}$, we trained a fully connected layer as the classifier Cls, mapping the high-dimensional features to the ovarian cancer subtype prediction results $s_{i}=Cls\left(f_{i}\right)$. We first used the Sigmoid function to map the prediction results to the classification probabilities, and then used cross-entropy as the loss function for classification. Since we used the fused features from all sequences, this is denoted as $L_{ALL}$: (2)$$L_{ALL}=E_{i}\left[y_{i}\cdot \log\left(\sigma \left(s_{i}\right)\right)+\left(1-y_{i}\right)\cdot \log\left(1-\sigma \left(s_{i}\right)\right)\right],$$
where $L_{ALL}$ represents the total loss of all samples in the fusion branch, ∑i denotes the summation over all samples, $y_{i}$ is the label indicating the ovarian cancer subtype of the patient, $s_{i}$ is the original prediction output for sample *i*, and $\sigma (s_{i})$ is the output of sample *i* after being passed through the sigmoid activation function σ. We also performed ovarian cancer subtype prediction for all single-sequence features, using a fully connected layer as the classifier and cross-entropy as the loss function, similar to the fused features. The sum of the losses from the single sequences and the multi-sequence led to the final loss *L*, which is: (3)$$L=L_{ALL}+L_{T1}+L_{T2},$$
where $L_{ALL}$ represents the total loss of all samples in the fusion branch, $L_{T1}$ represents the total loss of samples in the T1 branch, and $L_{T2}$ represents the total loss of samples in the T2 branch. The final loss function takes into account both the loss from the fused features and the loss from each individual branch.

### 2.7. Training and Implementation Details

During the training process, we used binary cross-entropy loss with logits as the loss function and employed AdamW as the optimizer to update the model parameters.The upsampling strategy employs weighted random sampling, where weights are calculated based on the number of samples in each category. The method for calculating weights involves taking the reciprocal of the number of samples in each category, meaning categories with fewer samples are assigned higher weights. Given $N_{0}$ and $N_{1}$ as the number of samples for class 0 and class 1 in the training class (train_cls), respectively, the formula for calculating weights is as follows:(4)$$\mathrm{weights}=\frac{1}{\left[\begin{array}{c} N_{0}\\ N_{1} \end{array}\right]}$$
where $N_{0}=\sum_{i=1}^{n}\left[train\_cls_{i}=0\right]$ and $N_{1}=\sum_{i=1}^{n}\left[train\_cls_{i}=1\right]$, where *n* is the total number of samples in the training class (train_cls), and $\left[train\_cls_{i}=0\right]$ and $\left[train\_cls_{i}=1\right]$ are indicator functions that equal 1 if $train\_cls_{i}$ is 0 or 1, respectively, and 0 otherwise. In mathematical terms, $\frac{1}{\left[\begin{array}{c} N_{0}\\ N_{1} \end{array}\right]}$ denotes taking the reciprocal of each element $N_{0}$ and $N_{1}$, yielding the weights for the respective classes. This method ensures that classes with fewer samples are assigned higher weights, thereby addressing the issue of data imbalance. In terms of data augmentation, a probability (*p*-value) of 0.3 was set, meaning that there was a 30% chance to select and execute a specific augmentation operation each time data augmentation was performed. This approach increases the diversity of the data while avoiding overfitting that might result from augmenting all images. The methods used include center cropping, scaling transformations, horizontal flipping, Gaussian noise, Gaussian smoothing, and contrast adjustment. The learning rate was initially set at 0.001, and it was gradually increased over the first 25 epochs. The learning rate was set to increase linearly as the ratio of the current epoch number to the number of warm-up epochs (25), and after the warm-up was completed (epoch greater than 25), the learning rate gradually decreased according to the cosine function [45]. The mathematical formula can be expressed as follows:

During the warm-up period (epoch ≤ warm-up_epoch): (5)$$\mathrm{lr}(\mathrm{epoch})=\frac{\mathrm{epoch}}{\mathrm{warm-up}\_\mathrm{epoch}}$$

At the beginning of training, during the first few epochs (complete passes through the dataset), the learning rate gradually increased from a lower value to a predetermined initial learning rate. This warm-up phase helped the model gradually adapt to the data in the early stages of training, avoiding instability in training that could be caused by setting the learning rate too high.

After the warm-up, the learning rate was adjusted using the cosine function (epoch > warm-up_epoch):(6)$$\mathrm{lr}(\mathrm{epoch})=0.5\times \left(\cos\left(\frac{\mathrm{epoch}-\mathrm{warm-up}\_\mathrm{epoch}}{\mathrm{num}\_\mathrm{epoch}-\mathrm{warm-up}\_\mathrm{epoch}}\times \pi \right)+1\right)$$

After the warm-up phase, the learning rate gradually decreased in the form of a cosine function until it approached zero. Cosine decay allows for the use of a larger learning rate in the early stages of training for rapid progress, while reducing the learning rate in the later stages to stabilize training. This avoids overly large parameter updates, thereby allowing for more fine-tuned adjustments of model parameters and achieving better training results. Here, lr(epoch) represents the learning rate for the current epoch, epoch represents the current training cycle, warm-up_epoch represents the number of epochs in the warm-up period, and num_epoch represents the total number of training epochs until training concludes. Our training process was conducted on the PyCharm platform using Python. The trained model was then applied to validation. Detailed parameter settings are shown in Table 2.

### 2.8. Metrics

The metrics used to measure model performance include AUC, AP, F1, ACC, SEN, and SPEC. The remaining experiments in this article use these six indicators as evaluation criteria.

•AUC (area under the curve): This is commonly used with the ROC (receiver operating characteristic) curve, referred to as ROC-AUC. The ROC curve is generated by plotting the true positive rate (TPR) against the false positive rate (FPR) for all possible classification thresholds. The AUC value is the area under this curve, ranging from 0 to 1. A higher AUC value indicates better model classification performance;•AP (average precision): This measures the average performance of the model’s precision (precision) across different thresholds. It is the area under the precision–recall curve, particularly suitable for evaluating imbalanced datasets. A higher AP indicates better model performance;•F1-Score (F1): This is the harmonic mean of precision (precision) and recall (recall). It is a number between 0 and 1 used to measure the model’s precision and robustness. A higher F1 score indicates a better balance between the model’s precision and recall;•ACC (accuracy): The most intuitive performance metric, indicating the proportion of correctly classified samples out of the total number of samples. A high accuracy means that the model can correctly classify more samples;•SEN (sensitivity) or recall: This is the true positive rate (TPR), measuring the model’s ability to correctly identify positive cases. A higher sensitivity means the model is more accurate in identifying positive cases;•SPEC (specificity): This is the true negative rate, measuring the model’s ability to correctly identify negative cases. A higher specificity means the model is more accurate in identifying negative cases.

## 3. Results

This study included 311 ovarian cancer (PAAD) patients, all of whom underwent two types of preoperative dynamic enhanced MRI examinations (T1WI+C and T2WI), with 212 patients having high-grade serous carcinoma (HGSOC) and 99 patients having clear cell carcinoma (OCCC). After random allocation, 198 patients were assigned to the training queue, another 50 patients formed the validation queue, and 63 patients were assigned to the testing queue. Moreover, the experiment employed 5-fold cross-validation to test the model’s robustness, with the results shown in Table 3.

EfficientNet-B2 was used as the feature extractor here, and the final result is the classification result of the fused features. To ensure the accuracy and reliability of the experimental results, a statistical analysis was performed on the results of five independently run experiments, calculating their mean and standard deviation. This method helps evaluate the stability and credibility of the model’s performance. The final statistical results are summarized in Table 4, showing the performance of our multi-parametric EfficientNet model across a range of key performance indicators. Specifically, the model achieved an average AUC value of 0.9162 on the test dataset, indicating its excellent ability to differentiate between positive and negative samples. At the same time, the model reached an average accuracy of 86.3% and an average F1 score of 0.8933, both of which reflect the model’s efficiency in correctly classifying samples. Furthermore, the average precision rate of 86.03% further confirms the model’s reliability in processing the test queue. The model also demonstrated an average sensitivity of 0.8651 and an average specificity of 0.85, meaning it can not only accurately identify positive cases but also effectively exclude negative ones. Notably, the standard deviation of all these metrics was below 0.1, highlighting the consistency and stability of the model’s performance.

Figure 4 provides a visual perspective to observe these results, showing the ROC (receiver operating characteristic) curves of the multi-parametric EfficientNet model utilizing fused features under five-fold cross-validation. The ROC curve is a tool for evaluating the performance of classification models by depicting the change in the true positive rate (TPR) against the false positive rate (FPR). In these five random splits of the experiment, except for the relatively weaker performance in the third fold (fold 3), the performance in other folds was excellent, with AUC values exceeding the 0.9 standard, further proving the strength and reliability of our model.

### 3.1. Impact of Feature Fusion on Results

Table 5 displays the prediction results of the T1 and T2 branches, separately, on the same test set. To ensure the fairness and comparability of the experiment, both branches were trained using settings identical to those identified as optimal in this study. The data points shown in the table were obtained by arithmetic averaging of the results from five independent experiments. This method helps reduce the impact of random variations, thereby providing a more stable and reliable performance assessment. It can be seen from Table 5 that the model proposed in this study outperformed the prediction results of the individual branches on almost all indicators. Optimal results are shown in bold.

### 3.2. Impact of Baseline Network Architecture on Results

Table 6 details the impact of different EfficientNet network architectures on the final prediction performance. Due to hardware limitations, EfficientNet-B5 to B7 were challenging to validate using the experimental setup, so only the effects of EfficientNet-B0 to B4 on the results were examined. To ensure a fair and direct comparison, all selected network structures were trained using the same parameter settings determined for the optimal model in this study, ensuring each model was evaluated under equivalent conditions. After conducting five independent experiments, the average prediction results for each network structure were calculated to obtain a robust performance evaluation. This method helps reduce the errors brought by the randomness of a single experiment, offering a more reliable and consistent performance measurement benchmark. The experimental results clearly demonstrated that the prediction model based on the EfficientNet-B2 architecture excelled across several key metrics among all the compared network structures. Therefore, EfficientNet-B2 was ultimately chosen as the baseline architecture for the model.

### 3.3. Impact of Hyperparameters on Results

In this section, we delve into the analysis of the relationship between model performance and its hyperparameter settings, especially focusing on how fine-tuning these parameters can achieve the best prediction effects. To comprehensively assess the impact of various hyperparameter configurations on model performance, we systematically varied four key parameters: the learning rate, fusion method, upsampling strategy, and cropping method. The experimental results are summarized in Table 7. Specifically, Table 7 records, in detail, the performance of the model across a range of performance metrics under different settings of learning rate, fusion method, upsampling, and cropping method. These performance metrics include AUC, AP, F1, ACC, SEN, and SPEC, forming a comprehensive framework for evaluating the model’s predictive capabilities. Through a comparative analysis of experimental results under different hyperparameter configurations, we discovered that, apart from the crop method, the choice of other hyperparameters had a very minor impact on the model’s results. The crop method, when set to CenterCrop, could significantly enhance the timing results. Choosing a learning rate of 0.001, a Concatenate fusion method, and employing upsampling offered a slight advantage in model performance. These constituted our final choice of hyperparameters.

## 4. Discussion

This article represents significant progress in the sub-classification of ovarian cancer. From an academic standpoint, we have developed a novel classification model utilizing multi-parametric MRI (mpMRI), achieving an AUC of 0.9162 and an accuracy of 89.33%. From a clinical perspective, we have made advancements in advising whether surgery or chemotherapy should be prioritized. Our research is specifically focused on the sub-classification of ovarian cancer, which contrasts with prior studies that primarily concentrated on the general classification of ovarian cancer, distinguishing tumor samples as either negative or positive [38,39,40,41,42]. Our approach holds greater clinical significance due to its ability to provide more nuanced categorization. We posit that integrating various parameters allows for a more comprehensive learning process within the model, thereby yielding favorable experimental outcomes.

Regarding limitations, several points are noteworthy: Firstly, while the sample size in this study suffices for exploratory scientific research and method validation [46,47], its expansion is warranted for generalization and broader clinical applicability. Combining a larger dataset with our model could enhance its potential clinical utility. Secondly, we are the first to undertake deep learning sub-classification research in this domain, and the absence of comparable studies limits our ability to benchmark our results [38,39,40,41,42]. Future investigations may involve comparative analyses with sub-classification efforts in other types of cancer. Lastly, substantial refinement is necessary before our model can be effectively deployed in clinical settings.

## 5. Conclusions

The model proposed in this article is feasible and plays a promotional role in both clinical application and scientific research. Our proposed model can sub-classify ovarian cancer, which is innovative compared to existing studies that categorize tumors as negative or positive, and there is a more urgent clinical demand for sub-classification. Overall, this study has potential for clinical application, and its limitations can be addressed through further research.

## Figures and Tables

**Figure 1 bioengineering-11-00472-f001:**
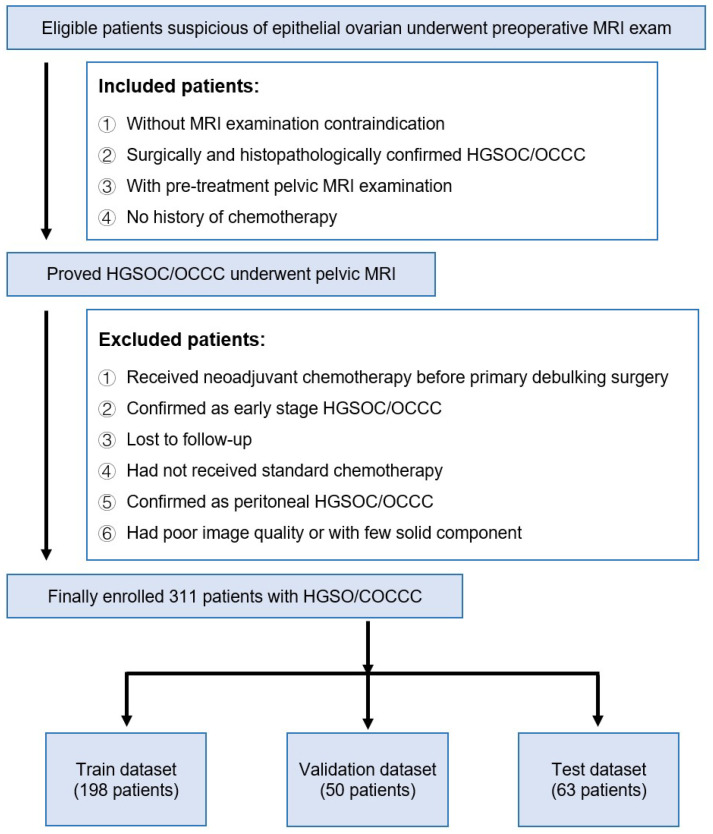
Flow chart for inclusion and exclusion of patients.

**Figure 2 bioengineering-11-00472-f002:**
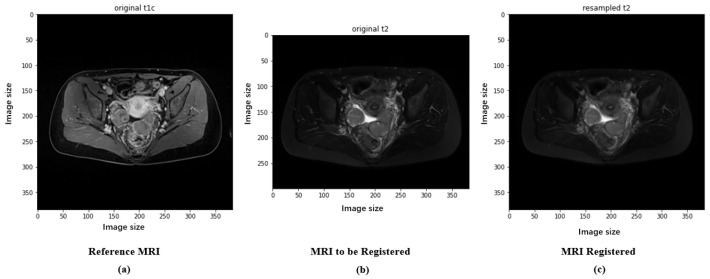
Images of experimental T1 and T2 data: (**a**) The original T1 image collected. (**b**) The original T2 image collected. (**c**) The registered T2 image.

**Figure 3 bioengineering-11-00472-f003:**
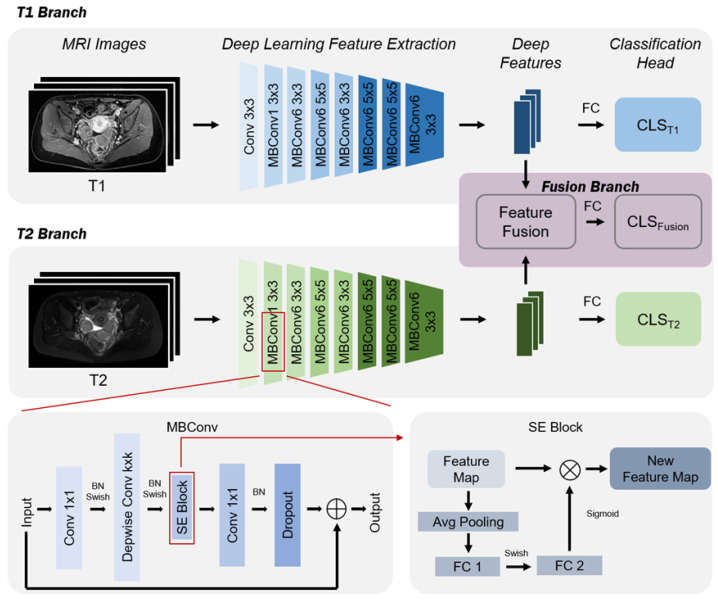
The multi-parameter fusion model architecture.

**Figure 4 bioengineering-11-00472-f004:**
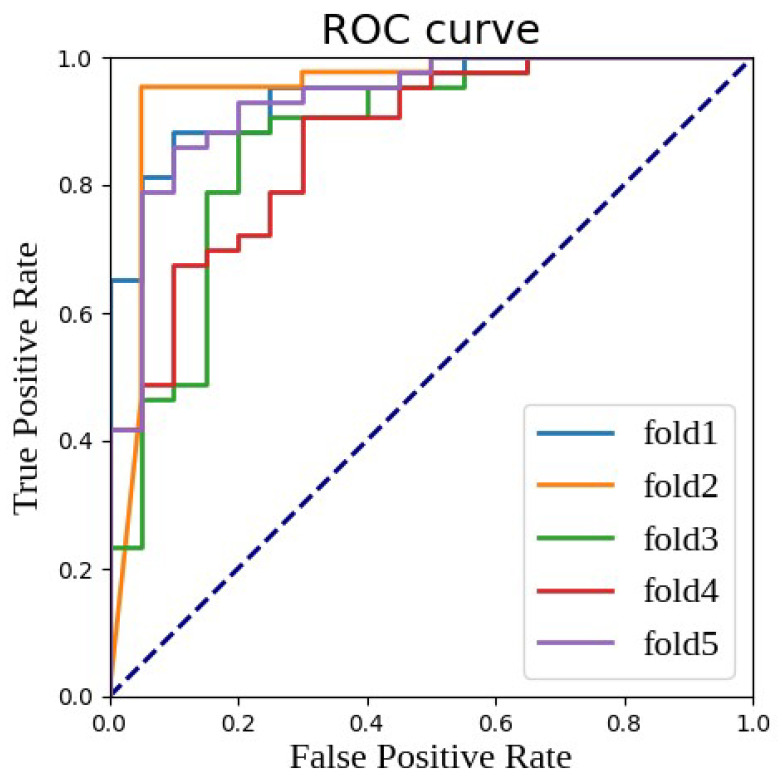
ROC curve and AUC value of fused features under different random divisions.

**Table 1 bioengineering-11-00472-t001:** Feature extractor network structure based on EfficientNet-B2.

Stage	Operator	Resolution	Channels	Layers
1	Conv3 × 3	224 × 224	32	1
2	MBConv1, k3 × 3	112 × 112	16	2
3	MBConv6, k3 × 3	112 × 112	24	3
4	MBConv6, k5 × 5	56 × 56	48	3
5	MBConv6, k3 × 3	28 × 28	88	4
6	MBConv6, k5 × 5	14 × 14	120	4
7	MBConv6, k5 × 5	14 × 14	208	5
8	MBConv6, k3 × 3	7 × 7	352	2

**Table 2 bioengineering-11-00472-t002:** Parameter Settings.

Hyperparameters	Description	Value
Learning Rate	Learning rate in model training	0.001
Batch Size	Number of samples per training batch	32
Number of Epochs	The total number of rounds of model training	100
Optimizer	Optimization algorithm used to train the model	AdamW
L2 Regularization	Only apply to the weight norm scale factors	0.0001
learning rate decay	Learning rate decay method with the number of training rounds	Warmup_Cosine(25)
Weight initialization	Initialization strategy for model weights	ImageNet Pretraining
Upsampling strategy	Dealing with imbalanced class distribution problems	Weighted Random Sampling
Data Augmentation Rate	Probability of applying data augmentation	0.3

**Table 3 bioengineering-11-00472-t003:** Five-fold cross-validation results.

	AUC	AP	F1-Score	ACC	SEN	SPEC
Fold 1	0.9442	0.9742	0.8916	0.8571	0.8605	0.8500
Fold 2	0.9413	0.9580	0.9302	0.9048	0.9302	0.8500
Fold 3	0.8814	0.9288	0.8989	0.8571	0.9302	0.7000
Fold 4	0.9116	0.9560	0.8889	0.8571	0.8372	0.9000
Fold 5	0.9023	0.9393	0.8571	0.8254	0.7674	0.9500

**Table 4 bioengineering-11-00472-t004:** Final test set results (mean ± standard deviation).

AUC	AP	F1-Score	ACC	SEN	SPEC
0.9162 ± 0.0226	0.9513 ± 0.0176	0.8933 ± 0.0261	0.8603 ± 0.0284	0.8651 ± 0.0686	0.850 ± 0.0935

**Table 5 bioengineering-11-00472-t005:** Comparison of T1 and T2 branch results and fusion branch results.

Branch	AUC	AP	F1-Score	ACC	SEN	SPEC
T1 Branch	0.7823	0.8615	0.8065	0.7524	0.7628	0.73
T1 Branch	0.8958	0.9472	0.8629	0.8286	0.8047	**0.88**
Fusion Branch	**0.9162**	**0.9513**	**0.8933**	**0.8603**	**0.8651**	0.85

**Table 6 bioengineering-11-00472-t006:** Comparison of the results using different baseline networks.

Baseline Network	AUC	AP	F1-Score	ACC	SEN	SPEC
EfficientNet-B0	0.8884	0.9216	**0.8941**	0.8571	0.8837	0.80
EfficientNet-B1	0.7663	0.8382	0.8913	0.8413	**0.9535**	0.63
EfficientNet-B2	**0.9162**	**0.9513**	0.8933	**0.8603**	0.8651	**0.85**
EfficientNet-B3	0.8215	0.9004	0.75	0.7143	0.6279	**0.85**
EfficientNet-B4	0.8291	0.909	0.8764	0.8254	0.907	0.65

Note: Bold data is the best for its column.

**Table 7 bioengineering-11-00472-t007:** Comparison of results using different hyperparameters.

Description	EfficientNet-B2					
Learning Rate	0.001	0.0001	0.001	0.001	0.001	0.001
Fusion Method	Concatenate	Concatenate	Add	Concatenate	Concatenate	Concatenate
Upsample	Yes	Yes	Yes	No	Yes	Yes
Crop Method	CenterCrop	CenterCrop	CenterCrop	CenterCrop	RandomCrop	Resize
AUC	**0.9162**	0.9093	0.9107	0.9009	0.6337	0.8756
AP	**0.9513**	0.9489	**0.9521**	**0.9566**	0.7874	0.9373
F1-Score	**0.8933**	0.8924	0.8571	**0.8991**	0.6389	0.881
ACC	0.8603	0.8467	0.8254	0.853	0.5873	0.8413
SEN	0.8651	**0.8902**	0.7674	0.9302	0.5349	0.8605
SPEC	0.85	0.78	**0.88**	0.73	0.70	0.80

Note: Bold data is the best for its column.

## Data Availability

Data presented in this study are not publicly available due to the ethical considerations.

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
