# Peer review of "Preoperative Molecular Subtype Classification Prediction of Ovarian Cancer Based on Multi-Parametric Magnetic Resonance Imaging Multi-Sequence Feature Fusion Network"

_bioengineering, 2024, doi:10.3390/bioengineering11050472_

Round 1
Reviewer 1 Report
Comments and Suggestions for Authors
The manuscript is devoted to a development of the deep neural networks for non-invasive assessment of preoperative classification of advanced serous carcinoma and clear cell carcinoma in ovarian cancer. The topic is important and corresponds to the aims and scope of the Bioengineering journal.
There are the following comments:
1. At the end of Introduction, it is necessary to add a description of the further structure of the manuscript.
2. The authors should more clearly present the contribution of this research to the subject area.
3. It is necessary to revise fully the structure of the sections of the manuscript. The manuscript should contain sections with statements of the research problem, the proposed methodology as well as NN architectures for solving it. Only then specific datasets and their processing with results obtained should be presented.
4. The division on train/test/validation datasets differs in the abstract and lines 92-93 as well as on Figure 1: 250 items for training and validation, 61 test ones versus 198+50+63.
5. Sections 2 and 3 are too short. It needs either to be expanded or merged.
6. Sections 4 and 5 in their current form should be merged.
7. In line 164, the value "1.010−3" should be rewritten in standard mathematical format.
8. The results section lacks a discussion of hyperparameters settings.
9. The manuscript does not compare the results with any other NN architectures, in particular, with the known state-of-the-art NN models for this problem. Unique dataset is definitely not enough here.
Author Response
Dear reviewer:
Thank you very much for giving us the opportunity to submit a revised version of the manuscript named "Preoperative Molecular Subtype Classification Prediction of Ovarian Cancer Based on mpMRI Multi-sequence Feature Fusion Network". We appreciate the time and efforts that you dedicated to our manuscript and are grateful for the valuable comments and suggestions. We have carefully considered all the comments and suggestions made by you, and we are uploading our response to the comments and concerns from reviewers, where italics indicate citations, blue italics indicate citations to review comments and black italics indicate citations to the original manuscript, red italics indicate citations to the newly revised manuscript.
Thank you and best regards,
Yijiang Du

Reviewer 2 Report
Comments and Suggestions for Authors
The article should be carefully revised.
The article does not fully reflect the results obtained and is characterized by careless presentation.

Minor editing of English language is required.
Author Response

(The authors gave the same response as above.)

Reviewer 3 Report
Comments and Suggestions for Authors
The article entitled “Preoperative Molecular Subtype Classification Prediction of Ovarian Cancer Based on mpMRI Multi-sequence Feature Fusion Network” is well-written and, from my point of view, would be of interest for the readers of Bioengineering.In spite of this, and before its publication, I would suggest authors to perform the following changes:
Introduction: a short description of the whole content (sections) of the manuscript would be of interest for readers.
Section 2: please discuss if the sample size is the adequate for this study.
Also, a section wih the foundation of the methodologies applied would also be of interest for readers.
Formulae 4 and 5 should be explained more in-depth and, from my point of view, also a reference would be of interest for a better understanding.
Table 4: expressing learning reate as 1E-4 is not acceptable. Please, express it as a power of 10. The same can be applied to Table 5, 6 and others…
From my point of view, a conclussions section should be mandatory in this research.
Author Response

(The authors gave the same response as above.)

Reviewer 4 Report
Comments and Suggestions for Authors
Title: "Preoperative Molecular Subtype Classification Prediction of
Ovarian Cancer Based on mpMRI Multi-sequence Feature
Fusion Network"
General comment: This work should be revised to improve the overall quality and impact. More specifically,
1. Introduction section:
*) This section could be enlarged and improved to account for the international contributions at the state of the art. Please rework.
*) Sections "2. Data Sources, Screening, and Division" and the following ones: it is currently lacking the standard "Materials and Methods" section: all the following sections belong to this section. In addition, also the final part of the current "Introduction" section (i.e., "In our study, we constructed a comprehensive clinical dataset, collecting data from 68
311 patients, of which 250 were used for training and validation, and 61 for independent 69
testing. All patients included enhanced T1 and T2 sequences, and the subtypes were 70
diagnosed by intraoperative pathological tissue biopsy. To evaluate the performance of the 71
built model, we used metrics such as the area under the receiver operating characteristic 72
(ROC) curve (AUC), accuracy, sensitivity, and specificity to assess the model’s efficacy." belongs to the "Materials and Methods" section.Please rework.
section "3. MRI Image Preprocessing"
*) This section is not totally clear, please rework and improve together with figure 2. In particular, it is not clear, what kind of images are used from the anatomical point of view. Are they totally omogenous or the cross sections of images are similar but not the same for each patient ? Please explain and improve. The fig.2 caption is not clear, please rework.
section "4. Ovarian Cancer Classification Prediction Model Based on mpMRI and EfficientNet Multi-sequence Feature Fusion "
Also this section should be a subsection of the "Material and Methods" section. Also this section is not clearly explained to the interested readers. The synergistic use of the T1 and T2 branches should be explained in all details. Please rework and improve (together with the caption of fig.3).
Lines: "EfficientNet has eight network architectures from B0 to B7, each differing in resolution, 124
channels, and layers[ 36]. Through experimentation, we used the convolutional part of 125
EfficientNet-B2 as the deep feature extractor, as shown in Table 1, where Conv3x3 represents 126
a conventional convolution layer with a kernel size of 3 × 3, MBConv1 indicates an expansion 127
factor of 1, MBConv6 indicates an expansion factor of 6, and k3×3 represents the kernel size 128
of 3 ×3 for layer-by-layer convolution within the respective module. It’s important to note 129
that the EfficientNet feature extractor weights for each sequence are different, with each 130
EfficientNet feature extractor extracting features only from the corresponding sequence. 131
We use the convolutional portion of EfficientNet-B2 as the deep feature extractor, 132
with its network structure as shown in Table 1. Here, Conv3x3 represents a conventional 133
convolution layer with a kernel size of 3×3, MBConv1 indicates a multiplier of 1 for 134
its expansion factor, MBConv6 indicates a multiplier of 6 for its expansion factor, and 135
k3× 3 represents the kernel size of 3×3 for layer-by-layer convolution in the corresponding 136
module. It is important to note that the weights of the EfficientNet feature extractor for 137
each sequence are different, with each EfficientNet feature extractor only extracting features 138
from the corresponding sequence. "
*) It is not clear how this experimental set-up has been selected. The authors should better explain their words "hrough experimentation, we used the convolutional part of 125
EfficientNet-B2 as the deep feature extractor, as shown in Table 1" and all the relevant details.
*) "4.2. Multi-Sequence Feature Fusion ": all the needed details should be provided to the interested readers. Please rework. Formula (1) should be better explained.
*) "4.3. Ovarian Cancer Subtype Prediction": formula 2 should be eplained in all details to the readers. Please improve.
Section "5. Training and Implementation Details"
*) This sections should be reorganized and inserted within the "Materials and Methods" section. Formulas 3 and 4 should be better explained in all the relevent details.
Section "6. Experimental Results"
*) This sections should be renamed "Results"
This section should be enlarged and improved. All the tables should better described together with Figure 4. Please enlarge and improve.
Section "Discussion".
*) This section should be enlarged and improved. In particular, the linitations of the current work should be better discussed with reference to the main results of this work.
Comments on the Quality of English Language
The quality of English could be improved
Author Response

(The authors gave the same response as above.)

Round 2
Reviewer 1 Report
Comments and Suggestions for Authors
All responses are clear. Regarding the comparison with the state-of-the-art NN models, this also meant independent testing of other NN architectures, not just direct comparison of the results obtained. This way you can avoid questions regarding non-public datasets. Nevertheless, this can be considered as a new independent research.
Author Response
Dear Reviewer,
Thank you very much for your recognition of our work. We are grateful for the time you have taken to review our manuscript again. Your ongoing attention and feedback are invaluable to us.
Best regards,
Yijiang Du
Reviewer 2 Report
Comments and Suggestions for Authors
The authors must correct the article in accordance with the comments.

Minor editing of English language is required.
Author Response
Dear reviewer:
Thank you very much for giving us the opportunity to submit a revised version of the manuscript "Preoperative Molecular Subtype Classification Prediction of Ovarian Cancer Based on mpMRI Multi-sequence Feature Fusion Network" again. We appreciate the time and efforts that you dedicated to our manuscript and are grateful for the valuable comments and suggestions. We have carefully considered all the comments and suggestions made by you, and we are uploading our response to the comments and concerns from reviewers, where italics indicate citations, blue italics indicate citations to review comments and black italics indicate citations to the original manuscript, red italics indicate citations to the newly revised manuscript

Reviewer 4 Report
Comments and Suggestions for Authors
Title: “Preoperative Molecular Subtype Classification Prediction of Ovarian Cancer Based on mpMRI Multi-sequence Feature Fusion Network “
Abstract: Using non-invasive methods to assess the preoperative classification of advanced serous carcinoma and clear cell carcinoma of ovarian cancer is an important and yet unresolved issue. Deep Learning methods based on the fusion of multiparametric magnetic resonance imaging have great potential for preoperative assessment of ovarian cancer classification. This article proposes a deep learning network architecture based on multi-sequence feature fusion to achieve high-accuracy prediction of preoperative advanced serous carcinoma and clear cell carcinoma classification. This study included a total of 311 patients, with 250 used for training and validation, and 61 for independent testing. All patients included enhanced T1 sequences and T2 sequences, and the classifications were determined by intraoperative pathological tissue biopsy. The performance of the proposed model was evaluated using the area under the ROC curve, accuracy, sensitivity, and specificity. The model proposed in this article achieved an average AUC of 0.9162, an average accuracy of 86.03%, an average sensitivity of 86.51%, and an average specificity of 85.0%. This article provides a new and effective approach for the non-invasive assessment of preoperative classification of advanced serous carcinoma and clear cell carcinoma in ovarian cancer.
General comment: The authors revised the manuscript. However, some major revisions should be further implemented to enhance the quality and the impact of this work.
Some specific comments:
Lines: “We believe that a sample size of 311 is sufficient for this study. Firstly, 113
public datasets on ovarian cancer subtypes are scarce, and the collection of clinical data is 114
very challenging. Secondly, in previously published literature, Zhang et al. [ 43 ] selected 115
299 patients, and Litjens et al. [44 ] chose 225 patients for their studies.”
*) Please move these lines to the “Discussion” section.
Lines: “In previous models, some have enhanced network performance by increasing the 162
network’s width, namely, the number of convolutional kernels (thus, increasing the chan- 163
nels of the feature matrix). This approach can achieve higher granularity of features and 164
also makes the network easier to train. However, networks that are significantly wide 165
but shallow often struggle to learn deeper-level features. Others have sought to enhance 166
network performance by increasing the network’s depth, i.e., using more layer structures, 167
which allows for the capture of more rich and complex features and can be well applied to 168
other tasks. However, an excessively deep network may encounter issues like vanishing 169
gradients and training difficulties. Some improve network performance by increasing 170
the input resolution; enhancing the image resolution of the network’s input can poten- 171
tially capture higher granularity feature templates, but for very high input resolutions, the 172
gains in accuracy also decrease. Moreover, high-resolution images increase computational 173
demands. 174
EfficientNet[ 41] utilized Neural Architecture Search (NAS) technology to find an 175
optimal configuration of three parameters: the network’s image input resolution, depth, 176
and channel width, to achieve higher predictive accuracy. This led to the development of 177
EfficientNet-B0 as the base model. Subsequently, through Compound Scaling, it systemati- 178
cally scaled the input image resolution, depth, and width of the base model, generating 179
a series of larger models: EfficientNetB1-B7. Through ablation studies, EfficientNet-B2 180
was ultimately chosen as the deep feature extractor. The part concerning ablation studies 181
will be discussed in detail in section four of the article”
*) These lines are mixed: please decouple the information about “Methods” and the comments about the used framework.
Lines: “2.5. Multi-Sequence Feature Fusion”, etc
*) This paragraph should be better explained and written. Please rework
Lines: The metrics used to measure model 275
performance include AUC, AP, F1, ACC, SEN, and SPEC, the remaining experiments in 276
this article use these six indicators as evaluation criteria: 277
• AUC (Area Under the Curve): Commonly used with the ROC (Receiver Operating 278
Characteristic) curve, referred to as ROC-AUC. The ROC curve is generated by plot- 279
ting the true positive rate (TPR) against the false positive rate (FPR) for all possible 280
classification thresholds. The AUC value is the area under this curve, ranging from 0 281
to 1. A higher AUC value indicates better model classification performance. 282
• AP (Average Precision): Measures the average performance of the model’s precision 283
(Precision) across different thresholds. It is the area under the Precision-Recall curve, 284
particularly suitable for evaluating imbalanced datasets. A higher AP indicates better 285
model performance. 286
• F1-Score (F1): The harmonic mean of precision (Precision) and recall (Recall). It is a 287
number between 0 and 1 used to measure the model’s precision and robustness. A 288
higher F1 score indicates a better balance between the model’s precision and recall. 289
• ACC (Accuracy): The most intuitive performance metric, indicating the proportion 290
of correctly classified samples out of the total number of samples. A high accuracy 291
means that the model can correctly classify more samples. 292
• SEN (Sensitivity) or Recall: The true positive rate (TPR), measuring the model’s 293
ability to correctly identify positive cases. Higher sensitivity means the model is more 294
accurate in identifying positive cases. 295
• SPEC (Specificity): The true negative rate, measuring the model’s ability to correctly 296
identify negative cases. Higher specificity means the model is more accurate in 297
identifying negative cases.
*) The description of the different kind metrics should be moved to the “Materials and Methods” section. Please rework accordingly.
“Discussion” and “Conclusions” sections could be improved.
References: The number and the quality of the cited journals should be improved.
Comments on the Quality of English Language
The quality of the English could be improved.
Author Response

(The authors gave the same response as above.)
